# Using Machine Learning Methods Combined with Vegetation Indices and Growth Indicators to Predict Seed Yield of *Bromus inermis*

**DOI:** 10.3390/plants13060773

**Published:** 2024-03-08

**Authors:** Chengming Ou, Zhicheng Jia, Shoujiang Sun, Jingyu Liu, Wen Ma, Juan Wang, Chunjiao Mi, Peisheng Mao

**Affiliations:** Forage Seed Laboratory, College of Grassland Science and Technology, China Agricultural University, Beijing 100193, China; b20203240983@cau.edu.cn (C.O.); b20193040362@cau.edu.cn (S.S.); 13789615104@163.com (J.L.);

**Keywords:** leaf nitrogen content, random forest model, remote vegetation index, seed yield prediction, smooth bromegrass

## Abstract

Smooth bromegrass (*Bromus inermis*) is a perennial, high-quality forage grass. However, its seed yield is influenced by agronomic practices, climatic conditions, and the growing year. The rapid and effective prediction of seed yield can assist growers in making informed production decisions and reducing agricultural risks. Our field trial design followed a completely randomized block design with four blocks and three nitrogen levels (0, 100, and 200 kg·N·ha^−1^) during 2022 and 2023. Data on the remote vegetation index (RVI), the normalized difference vegetation index (NDVI), the leaf nitrogen content (LNC), and the leaf area index (LAI) were collected at heading, anthesis, and milk stages. Multiple linear regression (MLR), support vector machine (SVM), and random forest (RF) regression models were utilized to predict seed yield. In 2022, the results indicated that nitrogen application provided a sufficiently large range of variation of seed yield (ranging from 45.79 to 379.45 kg ha⁻¹). Correlation analysis showed that the indices of the RVI, the NDVI, the LNC, and the LAI in 2022 presented significant positive correlation with seed yield, and the highest correlation coefficient was observed at the heading stage. The data from 2022 were utilized to formulate a predictive model for seed yield. The results suggested that utilizing data from the heading stage produced the best prediction performance. SVM and RF outperformed MLR in prediction, with RF demonstrating the highest performance (R^2^ = 0.75, RMSE = 51.93 kg ha^−1^, MAE = 29.43 kg ha^−1^, and MAPE = 0.17). Notably, the accuracy of predicting seed yield for the year 2023 using this model had decreased. Feature importance analysis of the RF model revealed that LNC was a crucial indicator for predicting smooth bromegrass seed yield. Further studies with an expanded dataset and integration of weather data are needed to improve the accuracy and generalizability of the model and adaptability for the growing year.

## 1. Introduction

Smooth bromegrass (*Bromus inermis*) is a perennial, cool-season rhizomatous forage grass of Eurasian origin known for its high nutritional value and tolerance to both drought and cold conditions [1]. It has extensive planting in pasture for ruminants and soil conservation [2,3]. However, as a perennial grass, its seed yield tends to gradually decline with increasing growing years [4]. Therefore, monitoring the growth status of the plant and accurately predicting seed yield are essential for growers to make agricultural decisions, optimize field management practices in advance, and reduce agricultural risks [5,6].

LAI and LNC are primary growth indicators for crop monitoring and yield prediction [7]. However, traditional destructive methods for measuring the biophysical and biochemical parameters of crops are time-consuming and labor-intensive [8]. With the advancement of remote sensing and spectral technologies, accurate estimation of crop growth indicators such as LAI and LNC can be achieved by combining vegetation indices and machine learning algorithms for regression modeling [9,10]. Additionally, vegetation indices like NDVI and RVI, reflecting the spectral reflection of the plant canopy, have been widely applied in vegetation cover density assessment [11], crop identification [12], and crop growth monitoring [13]. These indices have also found extensive applications in yield prediction for crops such as wheat (*Triticum aestivum*) [14], maize (*Zea mays*) [15], and cotton (*Gossypium*) [16].

Multiple linear regression (MLR) is a traditional statistical method that can predict the target variable by establishing a linear relationship between multiple variables and the target variable [17]. However, MLR can only explain linear relationships and its predictive results are not reliable for non-linear and more complex relationships [18,19]. With the advancement of artificial intelligence, machine learning improves predictions using multiple features, enhancing accuracy in non-linear and complex relationships [20,21]. Machine learning divides data into training and testing sets, using the former for modeling and training to establish non-linear relationships between independent and dependent variables. Subsequently, the model is evaluated using the testing set [21,22]. Random forest (RF) and support vector machines (SVM) are two commonly used machine learning algorithms that have been successfully applied to yield prediction in crops such as sugarcane (*Saccharum officinarum*) [23], rice (*Oryza sativa*) [24], maize [25], and wheat [26]. However, there was little research on the prediction of seed yield in perennial forage grasses.

This study employed traditional regression methods (MLR) and machine learning models (SVM and RF) to predict seed yield based on RVI, NDVI, LAI, and LNC. The objectives of this work were: (1) to evaluate performance of the three models on seed yield prediction of smooth bromegrass; and (2) to identify the optimal growth stage and indicator for predicting yield.

## 2. Results

### 2.1. The Explanatory Analysis of Seed Yield Components, Vegetation Index, and Growth Parameters

The seed yield in 2022 surpassed that of 2023, averaging 144.99 kg ha^−1^ and 50.42 kg ha⁻¹, respectively. In 2022, the seed yield ranged from 45.79 kg ha⁻¹ (0 kg·N·ha^−1^) to 379.45 kg ha⁻¹ (200 kg·N·ha^−1^) with an increase in nitrogen application. The seed yield of the CK treatment (0 kg·N·ha^−1^) was the lowest, while the seed yield of the N2 treatment (200 kg·N·ha^−1^) was the maximum. This increase was primarily attributed to the elevated values of FTS, SFT, FS, and SS. In 2023, nitrogen application (*p* < 0.05) increased FTS but had no significant impact on other yield components or seed yield (Table 1). Box plots showed the values of RVI, NDVI, LNC, and LAI for three different growth stages in both 2022 and 2023 (Figure 1). The values of RVI, NDVI, LNC, and LAI in 2022 exceeded those in 2023. Specifically, the average values in 2022 were 1.39 for RVI, 0.16 for NDVI, 0.9% for LNC, and 0.97 for LAI. Conversely, the average values in 2023 were 1.34 for RVI, 0.14 for NDVI, 0.78% for LNC, and 0.93 for LAI. Moreover, in 2022, nitrogen application significantly (*p* < 0.05) increased the values of RVI, NDVI, LNC, and LAI across the three growth stages. In 2023, nitrogen application significantly (*p* < 0.05) elevated the values of RVI, NDVI, LNC, and LAI at the anthesis stage and the milk stage. The growing years and different nitrogen application treatments provided a sufficiently large range of variability for the growth indicators (LAI and LNC), vegetation indices (RVI and NDVI), and seed yield.

### 2.2. Relationship between RVI, NDVI, LNC, LAI, and Seed Yield

In 2022, a significant (*p* < 0.01) positive correlation was observed among RVI, NDVI, LNC, LAI, and the seed yield across the three distinct growth stages, with RVI and LNC at heading stage having the highest correlation with seed yield (r = 0.867). Moreover, as the growth stages progressed, the correlation gradually decreased, with the maximum correlation coefficient observed at the heading stage. RVI, NDVI, LNC, and LAI showed a significant (*p* < 0.001) positive correlation, with correlation coefficients exceeding 0.96 (Figure 2). In 2023, there was no significant (*p* > 0.05) correlation between RVI, NDVI, LNC, LAI, and seed yield across the three growth stages. RVI, NDVI, LNC, and LAI exhibited a significant (*p* < 0.001) positive correlation, with the maximum correlation coefficient observed at the anthesis stage and the minimum correlation coefficient at the heading stage (Figure 3).

### 2.3. Performance Assessment of Machine Learning Models for Predicting Seed Yield Based on RVI, NDVI, LNC, and LAI

Based on the results of the correlation analysis, no significant correlations were observed between RVI, NDVI, LNC, LAI, and seed yield in 2023. Therefore, data from 2022 were used to formulate a predictive model for seed yield. The results revealed variations in model performance when predicting seed yield using data collected at the different growth stages (Table 2). Specifically, at the heading stage, the R^2^ values for MLR, SVM, and RF were 0.61, 0.72, and 0.75, respectively. The corresponding RMSE were 69.29, 57.39, and 51.93 kg ha^−1^. The MAE (29.43 kg ha^−1^) and MAPE (0.17) values of RF were both minimized. At the anthesis stage, the R^2^ values for MLR, SVM, and RF were 0.67, 0.64, and 0.63, respectively. The RMSE values were 59.69, 68.16, and 62.79 kg ha^−1^. The MAE (41.77 kg ha^−1^) and MAPE (0.31) values of SVM were both minimized. At the milk stage, the R^2^ values for MLR, SVM, and RF were 0.25, 0.59, and 0.59, respectively. The RMSE values were 109.53, 68.41, and 67.09 kg ha^−1^. The MAE (47.51 kg ha^−1^) value was minimized for SVM and the MAPE (0.30) value was minimized for RF. In conclusion, RF and SVM models outperformed MLR. Furthermore, these models showed optimal performance when utilizing data collected at the heading stage, with RF showing the most superior performance. Based on the results, the optimal model (RF) was used to predict the seed yield for 2023 but the predictive results were not significant (Appendix A). A variable importance analysis was performed for the RF model. The results revealed a shift in variable importance when utilizing data collected at the distinct growth stages for yield prediction. Notably, at the heading stage, the two most crucial variables were LNC and RVI (Figure 4A). At the anthesis stage, the top two important variables were NDVI and LNC (Figure 4B). At the milk stage, the top two important variables were RVI and LNC (Figure 4C). Remarkably, LNC emerged as the most important variable when utilizing data collected across all three growth stages (Figure 4D).

## 3. Discussion

The results demonstrated that machine learning models (SVM and RF) outperformed the traditional linear regression model (MLR) in predicting seed yield, consistent with previous studies [27,28]. This improvement might be attributed to the complex non-linear relationships between LAI, LNC, RVI, NDVI, and seed yield, which were not adequately complained by a simple linear model [29]. SVM and RF were better suited to interpret the complex relationships between input parameters and target variables [30,31], reducing overfitting risk and improving accuracy [32,33]. Furthermore, correlation results indicated that LAI, LNC, RVI, and NDVI at the heading stage exhibited the highest correlation with seed yield, rendering it the optimal period for yield prediction, with RF as the superior model (R^2^ = 0.75, RMSE = 51.93 kg ha^−1^, MAE = 29.43 kg ha^−1^, and MAPE = 0.17). RF possesses a notable advantage in assessing the importance of independent variables. Analysis of feature importance in RF models at different growth stages revealed varying critical features. Specifically, LNC was the most important at the heading stage, NDVI at the anthesis stage, and RVI at the milk stage. Combining data from all stages, LNC emerged as the most crucial feature, consistently ranking within the top two important features at the single stage. This indicated the significance of LNC in smooth bromegrass seed yield prediction. The RF model was used to predict the seed yield for the year 2023 using the data collected at the heading stage. The R^2^ (0.016) indicated poor performance of the model, potentially due to the low correlation between LAI, LNC, RVI, NDVI, and seed yield in 2023. Nonetheless, actual measurements showed that the seed yields for all treatments were quite low (<60 kg ha^−1^). And the model also predicted a relatively low seed yield for the year 2023 using heading stage data, with the predicted seed yield around 61 kg ha^−1^ and the RMSE value being 21.56 kg ha^−1^ (Appendix A). These results can serve as a reference basis for growers to make subsequent production and management decisions. However, the limitations of this study were possible overfitting due to the relatively small training and test datasets. Future research should focus on expanding the dataset to enhance the accuracy and generalizability of the RF model for smooth bromegrass seed yield prediction.

Many studies have reported growth indicators and vegetation indices as reliable predictors for yield prediction in various crops [7,16,34]. However, the prediction accuracy of seed yield varied when utilizing data collected at different growth stages. Previous studies have indicated a relationship between vegetation indices obtained at different crop growth stages and yield. Efforts have been made to enhance estimation accuracy using various modeling techniques [35,36]. The results of this study indicated that in 2022, LAI and LNC showed a significant positive correlation with seed yield (Figure 2), with the highest correlation coefficient being observed at the heading stage. The average values of LAI and LNC were also the highest at the heading stage, consistent with results observed in wheat [37]. This suggested that the heading stage represented the early stage of reproductive growth when leaf development was complete and the leaves possess higher photosynthetic capacity [38]. As growth stages progress, the values of LAI and LNC begin to decline, accompanied by a gradual decrease in correlation coefficients with seed yield. This may be attributed to leaf senescence occurring after the anthesis stage and the transfer of nitrogen from the leaves to the seeds [39,40]. Many studies have reported a strong correlation between NDVI and crop yield, especially during the flowering and grain-filling stages [41]. The results of this study also demonstrated that in 2022, RVI and NDVI showed a significant positive correlation with seed yield, with the highest correlation coefficient observed at the heading stage. However, as the growth stages progressed, the correlation coefficients gradually declined, which may be related to changes in leaf chlorophyll content and water content [42]. In conclusion, using data at the heading stage for smooth bromegrass seed yield prediction achieved the highest performance.

The seed yield of perennial forage grasses was influenced by factors such as precipitation, growing years, planting density, and fertilization [4,43]. In this study, significant decreases were observed in seed yield components and overall seed yield in the third year of growing. Previous study showed that with an increase in growing years, the seed yield of *Elymus kamoji* gradually decreased [44]. Additionally, the seed yield of slender wheatgrass (*E. trachycaulus*) and Siberian wildrye (*E. sibiricus*) began to decline after the second year of planting [43,44]. Furthermore, water deficiency at the nutrient growth stage might be a contributing factor to low seed yield. Given the rainfed management approach employed in this experiment, smooth bromegrass initiated regreening in late April, underwent nutrient growth in May and June, and entered the reproductive growth phase in July and August. The precipitation levels in May and June of 2023 were below the five-year average (2019–2023) (Appendix A), potentially resulting in nutrient deficiency, thereby affecting seed yield. The reduced values of RVI, NDVI, LNC, and LAI at the heading stage in 2023 also reflected the poor growth status of smooth bromegrass (Figure 1). Nitrogen was a crucial nutrient in the plant growth and development process, limiting crop growth, and reproduction. The application of an appropriate amount of nitrogen fertilizer can increase the number of fertile tillers and seeds per unit area, thereby enhancing the seed yield of annual crops such as wheat [45] and rice [46]. Reasonable nitrogen application had also significantly increased the seed yield of perennial forage grasses such as *Leymus chinensis* [47], *Lolium perenne* [48], and *Elymus sibiricus* [43]. In this study, the results showed that nitrogen application in 2022 significantly increased the seed yield components, thereby increasing the seed yield. However, in 2023, although nitrogen application increased FTS, it had no significant effect on seed yield. This may be attributed to the growing years and precipitation condition. The yield prediction model results (Table 2) suggested that during favorable growth conditions for smooth bromegrass (2022), data collected at the heading stage effectively predicted seed yield using the RF model. However, during years of poor growth conditions (2023), the model’s accuracy was expected to decrease. Therefore, in predicting seed yield for perennial forage grasses over multiple years, attention should also be given to interannual variations in climate conditions and the decline patterns of plants. 

## 4. Materials and Methods

### 4.1. Experimental Field

The experiment was conducted at the Forage Seed Production Experimental Base of the China Agricultural University during two growing seasons (2022–2023) at the Yuershan Ranch in Chengde city, Hebei province, China (41°44′ N, 116°8′ E; elevation 1455 m). The site falls within a semi-arid continental monsoon climate zone, experiencing an 85-day frost-free period. The monthly average temperatures and precipitation over the past five years (2019–2023) were shown in Appendix A: ERA5-Land monthly averaged data from 1981 to the present. Copernicus Climate Change Service (C3S) Climate Data Store (CDS)) (accessed on 15 January 2024). The soil characteristics of the trial included an organic matter content of 27.63 g·kg^−1^, available nitrogen of 20.58 mg·kg^−1^, available phosphorus of 10.40 mg·kg^−1^, and available potassium of 53.25 mg·kg^−1^. The experimental design followed a completely randomized block design with four blocks and three nitrogen application levels (0, 100, 200 kg·N·ha^−1^, denoted as CK, N1, and N2, respectively). Each plot measured 4 m × 5 m. Additionally, four untreated plots of 20 m^2^ each were randomly selected within the field and designated as T (without any field management). In all the field, smooth bromegrass seeds were sown on 8 July 2020, with a row spacing of 45 cm.

### 4.2. Measurement of RVI, NDVI, LNC, and LAI

The TOP-1200 plant canopy analyzer (Zhejiang Top Cloud Agriculture Technology Co., Ltd. Hangzhou, China.) was used to measure vegetation indices, including RVI and NDVI, as well as plant growth indicators LNC and LAI at the three growth stages (heading stage, anthesis stage, and milk stage). The calculation methods for RVI and NDVI were given by Equations (1) and (2). LAI and LNC were estimated using the model in the TOP-1200 plant canopy analyzer. The measurements were conducted on clear, calm mornings. Fifteen random points were selected in each plot and measurements were taken at a height of 0.5 m from the canopy. The specific measurement times were detailed in Table 3.
(1)$$\mathrm{RVI}\;=\frac{NIR}{R}$$
(2)$$\mathrm{NDVI}\;=\frac{NIR-R}{NIR+R}$$
where *NIR* and *R* represent spectral reflectance in near-infrared band and spectral reflectance in red wavelengths, respectively.

### 4.3. Measurement of Seed Yield and Yield Components

At the full maturity stage, sampling was conducted using 1 m segments as sampling units. One segment was randomly selected in each plot, uniformly harvested, placed in a mesh bag, transported to a drying area for air-drying (at 20–25 °C, for 2–3 days), and subsequently manually threshed and cleaned. Seeds were weighed using a percentage scale and converted to seed yield (kg ha^−1^). Additionally, one randomly selected 1 m segment within each plot was used to determine the number of fertile tillers and converted to fertile tillers m^−2^ (FTS). Within the sampled row, 30 fertile tillers were randomly chosen to count the number of spikelets per fertile tiller (SFT). From the previously selected 30 fertile tillers, an additional 30 spikelets were randomly chosen to count the number of florets per spikelet (FS) and seeds per spikelet (SS).

### 4.4. Seed Yield Prediction Methods

Multiple linear regression (MLR), support vector machine (SVM), and random forest (RF) methods were conducted to predict the seed yield. MLR was a traditional linear regression method widely employed in yield prediction [17]. MLR utilized multiple variables to predict the target variable, resulting in predictions of greater accuracy compared to univariate forecasting. This was because changes in a dependent variable were often associated with variations in multiple independent variables [49]. In this study, stepwise regression was used to mitigate the impact of multicollinearity. SVM was a supervised learning model used for classification and regression analysis. SVR was capable of handling non-linear relationships by mapping data into a higher-dimensional space for better fitting of complex relationships. SVR included four different kernel functions: linear, polynomial, spline, and radial [50]. In this study, SVM with a radial kernel was chosen because it demonstrated the best performance in yield prediction among the four kernel functions. RF was an ensemble learning method that enhanced model accuracy through integrating multiple decision trees [32]. RF was effective in evaluating the importance of independent variables, addressing multicollinearity issues, and exhibiting strong resistance to overfitting with excellent generalization capabilities [33,51].

The data were split, allocating 70% for training and reserving 30% for testing, with five-fold cross-validation being conducted. The coefficient of determination (R^2^), root–mean–square error (RMSE), mean absolute error (MAE), and mean absolute percentage error (MAPE) were used to assess the model performance.
(3)$$R^{2}=1-\frac{\sum_{i=1}^{n}{\left(T_{i}-P_{i}\right)}^{2}}{\sum_{i=1}^{n}{{(T}_{i}-\bar{T})}^{2}}$$
(4)$$\mathrm{RMSE}=\sqrt{\frac{1}{n}\sum_{1}^{n}{\left(P_{i}-T_{i}\right)}^{2}}$$
(5)$$\mathrm{MAE}=\frac{\sum_{i-1}^{n}\left|P_{i}-T_{i}\right|}{n}$$
(6)$$\mathrm{MAPE}=\frac{1}{n}\sum_{i=1}^{n}\left|\frac{P_{i}-T_{i}}{T_{i}}\right|$$

### 4.5. Data Analysis

Duncan’s test was employed to examine seed yield and the values of RVI, NDVI, LAI, and LNC at different nitrogen levels, with a significance threshold of *p* < 0.05. The Student’s *t*-test (*p* < 0.05) was employed to examine seed yield and the values of RVI, NDVI, LAI, and LNC between 2022 and 2023. The R programming language (R 4.1) was used to create plots.

## 5. Conclusions

Accurately predicting seed yield is essential to assist growers in making informed production decisions. This study found that the traditional regression method (MLR) and machine learning models (SVM and RF) combined with RVI, NDVI, LAI, and LNC can predict seed yield for smooth bromegrass. Furthermore, the RF model demonstrated the highest predictive performance by using the data at the heading stage for seed yield prediction. LNC emerged as a crucial indicator for predicting smooth bromegrass seed yield. However, the comparatively small size of the training and test datasets in this study might impede the accuracy and generalizability of the model. In addition, the prediction accuracy of seed yield for perennial grass was influenced by growing years and weather conditions. Therefore, in future research, it is advisable to expand the dataset and incorporate meteorological data into the model development process.

## Figures and Tables

**Figure 1 plants-13-00773-f001:**
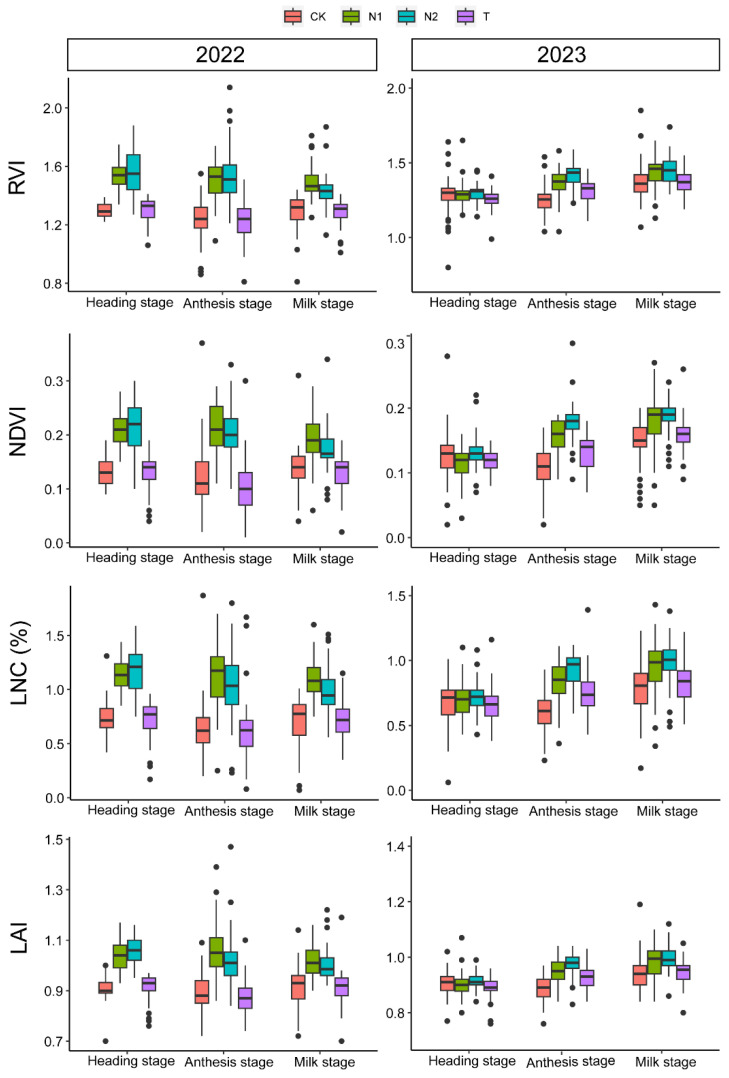
The effect of nitrogen and the growing year on RVI, NDVI, LNC, and LAI. CK: 0 kg·N·ha^−1^; N1: 100 kg·N·ha^−1^; N2: 200 kg·N·ha^−1^; T: plot without any field management.

**Figure 2 plants-13-00773-f002:**
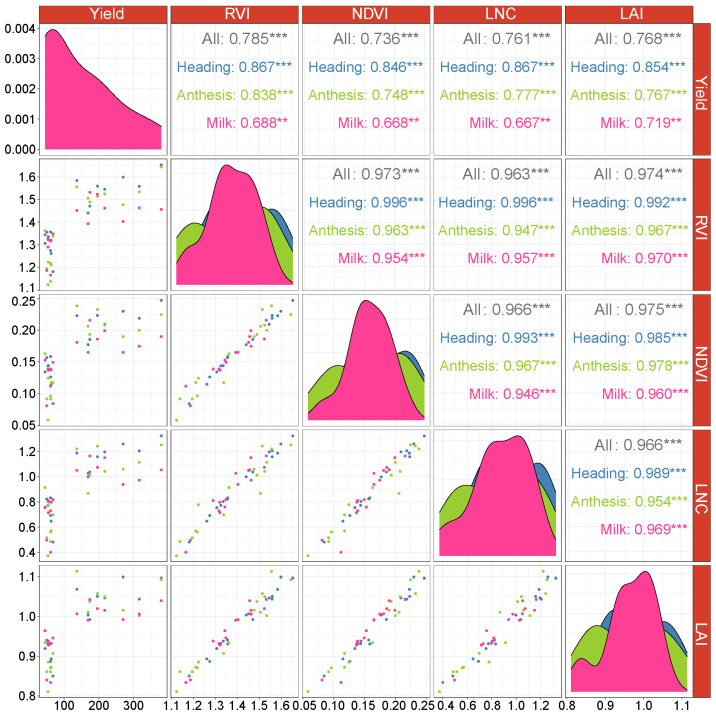
Pearson correlation analysis between RVI, NDVI, LNC, LAI, and seed yield in 2022: ** significant at the 0.01 probability level; *** significant at the 0.001 probability level. Bule means data collected at heading stage; green means data collected at anthesis stage; pink means data collected at Milk stage.

**Figure 3 plants-13-00773-f003:**
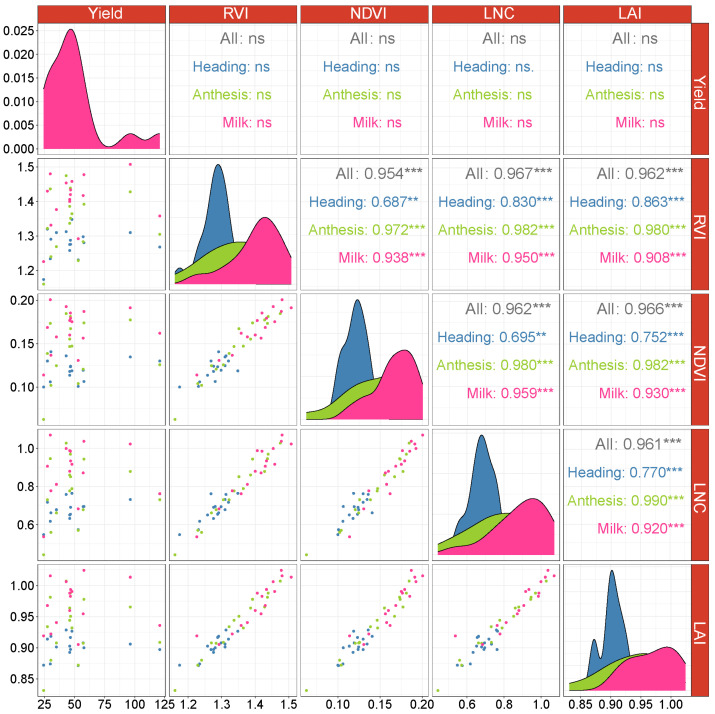
Pearson correlation analysis between RVI, NDVI, LNC, LAI, and seed yield in 2022: ** significant at the 0.01 probability level; *** significant at the 0.001 probability level; ns: not significant. Bule means data collected at heading stage; green means data collected at anthesis stage; pink means data collected at Milk stage.

**Figure 4 plants-13-00773-f004:**
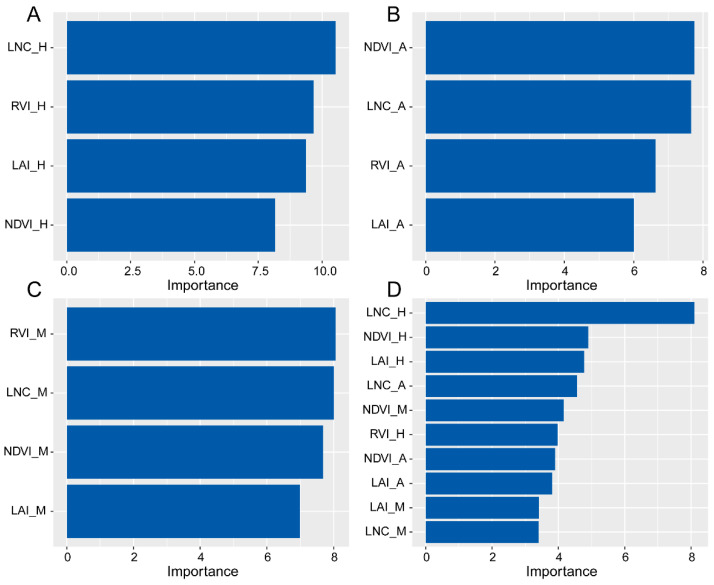
The *p* variable importance of RF per growth stages: (**A**): heading stage; (**B**): anthesis stage; (**C**): milk stage; (**D**): all three growth stages. LAI_H, LNC_H, NDVI_H, and RVI_H represent the data for LAI, LNC, NDVI, and RVI collected at heading stage; LAI_A, LNC_A, NDVI_A, and RVI_A represent the data for LAI, LNC, NDVI, and RVI collected at anthesis stage; LAI_M, LNC_M, NDVI_M, and RVI_M represent the data for LAI, LNC, NDVI, and RVI collected at milk stage.

**Table 1 plants-13-00773-t001:** The effect of nitrogen and the growing year in seed yield and yield components.

Year	Treatment	Fertile Tillers m^−2^ (FTS)	Spikelets per Fertile Tiller (SFT)	Florets per Spikelet (FS)	Seeds per Spikelet (SS)	Seed Yield (kg ha^−1^)
2022	CK	34.26 ± 2.74 c	21.52 ± 0.81 b	4.63 ± 0.07 bc	2.78 ± 0.1 b	58.07 ± 4.11 c
	N1	61.11 ± 3.12 b	27.89 ± 1.07 a	5.46 ± 0.27 a	3.51 ± 0.14 a	181.01 ± 17.09 b
	N2	97.78 ± 7.1 a	31.22 ± 1.6 a	4.85 ± 0.2 ab	3.26 ± 0.23 ab	283.7 ± 44.23 a
	T	30.74 ± 2.13 c	21.07 ± 0.71 b	4.13 ± 0.21 c	2.75 ± 0.24 b	57.19 ± 3.91 c
2023	CK	35.56 ± 2.4 b	16.11 ± 1.04 a	4.3 ± 0.17 ab	1.8 ± 0.04 a	34.7 ± 7.57 a
	N1	67.41 ± 7.28 a	16.44 ± 1.96 a	4.7 ± 0.14 a	2.08 ± 0.16 a	42.49 ± 6.33 a
	N2	76.3 ± 2.77 a	14.08 ± 0.63 a	4.53 ± 0.19 ab	1.81 ± 0.25 a	45.33 ± 0.86 a
	T	46.11 ± 4.66 b	14.06 ± 0.26 a	4.08 ± 0.09 b	1.68 ± 0.05 a	48.24 ± 1.67 a

Note: different letters are significantly different at the 0.05 level. CK: 0 kg·N·ha^−1^; N1: 100 kg·N·ha^−1^; N2: 200 kg·N·ha^−1^; T: plot without any field management.

**Table 2 plants-13-00773-t002:** The accuracy evaluation prediction results at the three growth stages.

Stage	ML	R^2^	RMSE	MAE	MAPE	*p*_Value
Heading stage in 2022	MLR	0.61	69.29	56.74	0.48	<0.01
	SVM	0.72	57.39	38.47	0.26	<0.01
	RF	0.75	51.93	29.43	0.17	<0.01
Anthesis stage in 2022	MLR	0.67	59.69	45.48	0.50	<0.01
	SVM	0.64	68.16	41.77	0.31	<0.01
	RF	0.63	62.67	43.64	0.35	<0.01
Milk stage in 2022	MLR	0.25	109.53	82.29	1.01	<0.01
	SVM	0.59	68.41	47.51	0.34	<0.01
	RF	0.59	67.09	48.89	0.30	<0.05

Note: R^2^: coefficient of determination; RMSE: root–mean–square error; MAE: mean absolute error; MAPE: mean absolute percentage error; *p*_value: significance of the predictive model.

**Table 3 plants-13-00773-t003:** Dates for RVI, NDVI, LNC, and LAI measurements.

Stage	2022	2023
Heading stage	7 July 2022	6 July 2023
Anthesis stage	14 July 2022	19 July 2023
Milk stage	31 July 2022	3 August 2023

## Data Availability

Data are contained within the article and Appendix A.

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
