# Peer review of "Using Machine Learning Methods Combined with Vegetation Indices and Growth Indicators to Predict Seed Yield of Bromus inermis"

_plants, 2024, doi:10.3390/plants13060773_

Round 1

Reviewer 1 Report

Comments and Suggestions for Authors

Journal: Plants (ISSN 2223-7747)

Manuscript ID: plants-2875763

Manuscript Title: "Using Machine Learning Methods Combined with Vegetation Indices and Growth Indicators to Predict Seed Yield of Smooth Bromegrass”

The manuscript is within the scope of the journal and the results are interesting, with potential of practical use and the manuscript is well prepared. However, there are some major concerns in respect to the current version of manuscript. The specific comments are as follows:

1-  In the title L3-4: “Smooth Bromegrass” It is preferable to write the scientific name (Bromus inermis) instead of the common name.

2-  L10: “growing year” what do the authors mean? Growing season???

3-  It is preferable for researchers to write a brief summary of the statistical design of the experiment in the abstract section.

4-  In the abstract section, methodology should be mentioned with more clarification.

5-  L15-16: The abstract part lacks the presence of numerical data or percentages of the level of increase and decrease as an important result.

6- L21, L119, L171, L298: “We” ……, L209, L227, L311: “our” Personal pronouns should not be used throughout the manuscript, avoid writing them out.

7-  It should be written a recommendation for this study in the form of short sentences at the end of the abstract part.

8- Keywords: Pls write the keywords in alphabetical order. Pls provide significant words which are more relevant to the work in a logical sequence as "keywords". Rephrasing some words, do not repeat as key files mentioned in the title of the current study. Pls capitalize the first letter of each word.

9-  L 40: “Leaf Nitrogen Content (LNC)” All terms that were previously abbreviated, only their abbreviations are written throughout the manuscript. Please pay attention to this comment throughout the manuscript.

10-L 46-47: “Normalized Difference Vegetation Index (NDVI) and Remote Vegetation Index (RVI)” Same previous comment.

11-  Hypothesis of the work should be clear at the end of the introduction part.

12-  L75: “In 2022, the seed yield ranged from 45.79 to 379.45 kg ha⁻¹”. Pls determine the nitrogen treatments that led to these increases.

13- L 77: “N2” The first appearance of abbreviations needs to be marked with complete definitions. I know that these experimental treatments are mentioned in detail in the materials part, but according to the journal instructions, the materials part comes after the discussion part.

14- L74-75: “The seed yield in 2022 surpassed that of 2023, averaging 144.99 kg ha⁻¹ and 50.42 kg ha⁻¹, respectively.” What is the cause of this great disparity in seed yield? Please provide convincing explanations for this in the discussion section

15-L77-79: “This increase was primarily attributed to the elevated values of FTS, SFT, FS and SS. In 2023, nitrogen application (P < 0.05) increased FTS but had no significant impact on other yield components and seed yield (Figure 1)” Unfortunately, it is very difficult for the Figure reader to determine the significance of the results or not through Figure, as there is nothing to explain this. The authors could have simplified drawing the figures in a clear way that could be easily understood. Kindly improve.

16- I noticed that "smooth bromegrass "was repeated 31 times throughout the manuscript (6 times in the abstract only). I believe that this is an excessive use of the name of the plant under study. I suggest reducing its writing except in necessary places only because this weakens scientific writing. Pls review similar instances of other terms throughout the manuscript.

17-L109-111: “Figure 3. Pearson correlation analysis between RVI, NDVI, LNC, LAI and seed yield in 2022. *Significant at the 0.05 probability level; **Significant at the 0.01 probability level; ***Significant at the 0.001 probability level; ns, no significant.” Pls remove “ns, no significant” Because there is no need to write it for this page.

18- In figure (4), pls write “ns, not significant” instead of mentioned values in the top row of this table

19-In “Table (1) The accuracy evaluation prediction results at three growth stages” pls explain, why did the two studies not include results for Anthesis Stage in 2022, Milk Stage in 2022 and All in 2022 for the year 2023?

20-In Table (1), please also include evidence of the significance of the results or not, because there is no evidence of this in the table values.

21-All sub-headings must be removed from the discussion part, where the discussion part must be one unit, in which the studied characteristics are discussed and ends with an adequate summary of everything studied.

22-L 173-178: “Nonetheless, Actual measurements presented that the seed yield for all treatments were quite low ………..production and management decisions. I have a very important question in this regard. It is known that adding nitrogen fertilizers positively affects the plant growth and thus the yield in most cases, through its main role in protein production and other jobs that are known about the role of nitrogen and its functions for plants as mentioned in Lines 222-232 in the current study. If field measurements and actual crop measurements are taken, the yield will increase significantly compared to the control. But by predictions like our study, the yield did not increase significantly as mentioned in Lines 209-211. How can authors explain this?

23-   L214-215: “Furthermore, nutrient deficiency may also be a contributing factor to low seed yield.” But the authors added nutrient fertilizers in the current study.

24- L “Each plot measured 4 m × 5 m. Additionally, four untreated plots of 20 m2 each were randomly selected within the field and designated as T. Smooth bromegrass seeds were sown on July 8, 2020, with a row spacing of 45 cm” pls this sentence needs more explanation. Kindly explain.

25- In the conclusion, L308-310: “In this experiment, the planting area for smooth bromegrass was located in a semi-arid region, and rain-fed management practices were implemented. The seed yield was influenced by the growing year and climatic conditions.” Pls remove this sentence, because its meaning was repeated or repeated more than once, especially in the abstract and introduction.

26- You focused only on the study strengthens and ignored the limitations, please discuss the study limitations in the conclusion.

27-  On the other hand, please mention how the future study can complete your work. What is the lack of knowledge?

28-  L452: “Leymus Chinensis” the scientific names have to be italic in the entire manuscript.

29-   General comment; Pls, shorten the long sentences as soon as possible and avoid repetition

30-  Besides the previous suggestion, a thorough proof-check (punctuation, spelling/typing) and most importantly English language is recommended.

Comments on the Quality of English Language

Minor editing of English language required

Author Response

Dear Reviewer,

Thank you very much for your responsible and professional suggestions. According to your comments and suggestions, we modified the whole revision.

Thanks a lot.

Kind regards,

Peisheng Mao

Reviewer 2 Report

Comments and Suggestions for Authors

The research presented in the manuscript is very interesting. However, the manuscript should be revised and supplemented with certain information. Figure 1 should be prepared in the same style as Figure 2.

Abbreviations of experimental combinations on the graphs should be explained in the caption.

Combination T should be better explained and justified, whether it was a field only with soil (or if there were any weeds growing, for example).

What does "corr" mean in Figure 3 and 4?

Why are only the data for 2022 presented in Table 1?

Figure 5 lacks explanations for the abbreviations on the y-axis. W

hen using all variables in the model, aren't the authors concerned about multicollinearity?

There is a lack of explained abbreviations for the formulas used to evaluate the models.

Author Response

(The authors gave the same response as above.)

Round 2

Reviewer 1 Report

Comments and Suggestions for Authors

Journal: Plants (ISSN 2223-7747)-R2

Manuscript ID: plants-2875763

Manuscript Title: "Using Machine Learning Methods Combined with Vegetation Indices and Growth Indicators to Predict Seed Yield of Smooth Bromegrass”

After carefully reviewing the entire corrected manuscript and after reviewing the extent of the researchers' response to my previous comments on it, I found that the authors had responded well to the comments and that the manuscript had greatly improved, making it suitable for publication in the Plants after minor modifications. The specific comments are as follows:

1-  Keywords: Pls write the keywords in alphabetical order.

2-      L75: “In 2022, the seed yield ranged from 45.79 to 379.45 kg ha⁻¹”. Pls determine the nitrogen treatments that led to these increases.

3-      L109-111: “Figure 3. Pearson correlation analysis between RVI, NDVI, LNC, LAI and seed yield in 2022. *Significant at the 0.05 probability level; **Significant at the 0.01 probability level; ***Significant at the 0.001 probability level; ns, no significant.” Pls remove “ns, no significant” Because there is no need to write it for this page.

Comments on the Quality of English Language

Minor editing of English language required

Author Response

Dear Reviewer,

Thank you very much for your responsible and professional suggestions. According to your comments and suggestions, we have modified the revision. The responses are shown as followed.

1-  Keywords: Pls write the keywords in alphabetical order.

Response: We have made the modification in the revision. 

2-      L75: “In 2022, the seed yield ranged from 45.79 to 379.45 kg ha⁻¹”. Pls determine the nitrogen treatments that led to these increases.

Response: We have made the modification in the revision. "In 2022, the seed yield ranged from 45.79 kg ha⁻¹ (0 kg·N·ha-1) to 379.45 kg ha⁻¹ (200 kg·N·ha-1) with the increase of the nitrogen application. The seed yield of the CK treatment (0 kg·N·ha-1) was the lowest, while the seed yield of the N2 treatment (200 kg·N·ha-1) was the maximum."

3-      L109-111: “Figure 3. Pearson correlation analysis between RVI, NDVI, LNC, LAI and seed yield in 2022. *Significant at the 0.05 probability level; **Significant at the 0.01 probability level; ***Significant at the 0.001 probability level; ns, no significant.” Pls remove “ns, no significant” Because there is no need to write it for this page.

Response: L109-111: In Figure 2, we have removed "ns, no significant".

Thanks a lot.

Kind regards,

Peisheng Mao

Reviewer 2 Report

Comments and Suggestions for Authors

The authors significantly improved the manuscript, taking into account my suggestions

Author Response

Dear Reviewer,

Thank you very much again for your responsible and professional suggestions. 

Thanks a lot.

Kind regards,

Peisheng Mao